# Motivational Typology of Online Food Souvenir Shoppers and Their Travel-Related Intentions

**Shu-Chun Lucy Huang \*** , **Chih-Yung Wang and Yi-Ru Yan**

Department of Tourism, Shih Hsin University, Taipei 116, Taiwan; cywang@mail.shu.edu.tw (C.-Y.W.);
juliej840615@gmail.com (Y.-R.Y.)
**\*** Correspondence: huangsch@cc.shu.edu.tw

**Abstract:** Online shopping has tremendous growth nowadays. Concerns about whether people perceive food souvenirs as commercial goods or products with local connections arise when shopping online for those souvenirs. Another concern is whether people remain interested in knowing or visiting places where food souvenirs originate even if they can simply purchase food souvenirs online. These issues are important for the sustainable development of local tourism. This study aims to investigate consumers' motivations for purchasing food souvenirs online and whether differences exist between segmented consumers in relation to their travel-related intentions. Participants of this work are individuals who have purchased food souvenirs online and are aged 18 years old or above. Social media applications, such as Facebook and Line, were utilized as the platforms for the questionnaire survey. Participants' motivations for purchasing food souvenirs online included five factors: exploring trends, local affiliations, social interactions, frugal sampling, and enforcing relationships. According to their motivations, participants were segmented into four types of consumers, including pleasure reminiscers (47.0%), journey recallers (22.9%), economical tasters (17.4%), and social practicers (12.7%). Pleasure reminiscers have stronger intentions toward searching for information on, traveling to, and recommending the places where food souvenirs originate compared with the other three types of consumers. On the basis of the findings, suggestions for the design and marketing of food souvenirs are provided for the producers of food souvenirs and promoters of local tourism.

**Keywords:** food souvenir; online shopping; shopping motivation; shopper typology; tourism development

## 1. Introduction

In Taiwan, "Bonsholi" refers to food souvenirs purchased by travelers as a gift for family and friends [1], for which "Omiyage" is the Japanese equivalent. Traditionally, Bonsholi is a gift given to strengthen interrelationship connections. Nowadays, Bonsholi is the term used by tourist destinations to market local food souvenirs [2]. Snacks made of local agricultural produce are often the favorite purchase of tourists when they buy souvenirs [3]. A survey [4] among domestic tourists shows that buying food souvenirs and trying the local gastronomy (42.8%) ranked as the second most popular activity during travel. Shopping is believed to promote tourist satisfaction. Furthermore, tourists' purchases contribute to the economy of their destinations and help in achieving the goals of sustainable tourism [5]. Food souvenirs are particularly produced in connection with elements of local culture or events to encourage bulk purchases and signify symbolic meanings that cannot be ignored [3]. Tourists' interest in local produce may serve to stimulate local awareness and curiosity, encourage community pride, and facilitate reinforcement of local identity and culture [6]. Food sales contribute to the destination economy, and the income from food souvenirs is beneficial for the environmental and cultural sustainability of tourism destinations [7–9].

Generally, food souvenirs in Taiwan can be purchased from stores in tourist destinations or through marketing distribution channels, such as agricultural fairs, convenience stores, or supermarkets. In recent years, many online shopping platforms began to sell food souvenirs and have become competitive in relation to other sales channels. Online shopping is now one of the most popular online activities worldwide [10] and provides consumers with a means to escape boredom, entertain themselves, and experience enjoyment [11]. In Taiwan, an annual survey of 1624 participants aged 15 to 64 years reveals that the percentage of online shopping (85.6%) has exceeded that of shopping at brick-and-mortar stores (77.7%) for the first time in 2018 [12]. The proliferation of online shopping has gained substantial attention among businessmen and researchers. Past studies have identified the primary motives for online shopping, including overall savings, convenience, information seeking, social interaction, and shopping experience [13]. The benefits of online shopping for consumers include convenience, variety of selection, low prices, original services, personal attention, and easy access to information [14].

Souvenirs are material objects that link people with places and memories [15]. Food souvenirs are especially more grounded in the culture, geography, and climate of a destination in comparison to non-food counterparts, such as t-shirts and cups [16]. Nowadays, online retail has achieved tremendous growth. Consequently, whether people perceive food souvenirs as ordinary commercial goods or products with local connections has become an issue when shopping online for food souvenirs. For regular commercial edible goods, the places where the material and production originate are not of great concern to most consumers. For food souvenirs, however, the material and production present the geographical and sociocultural uniqueness of a specific locale and are full of local meanings [3]. When people's motivations for purchasing food souvenirs are linked to local significance, local food is likely to become a driver for tourists' visits. Therefore, knowing the motivational segments of online food souvenir shoppers allows destination promoters and retailers to better understand the expectations and desires of their consumers. Another concern is whether people remain interested in knowing or visiting places where food souvenirs originate even if they can simply purchase food souvenirs online. Prior research confirmed that when local food becomes an attraction to tourists, this development increases the circulation of tourist expenditure through the local economy and is beneficial for sustaining local tourism development [7]. Accordingly, knowing how to make food souvenirs iconic and stimulating online shoppers' desire to visit the locale where food souvenirs originate is important for the development of local tourism.

Past studies on souvenir shopping during travel mainly investigated brick-and-mortar stores [17–23]. These investigations explored the tourists' motivation for shopping [19,20], and souvenir shoppers have been segmented [23]. However, no research on food souvenir shopping in an online context has been found. This study seeks to investigate consumers' motivations for online purchase of food souvenirs and to segment consumers according to their motivations. Moreover, attempts are made to examine whether differences exist between segmented consumers as regards their intentions of travel-related behaviors, including willingness to search for information on, traveling to, and recommending places where food souvenirs originate. The findings of this study are beneficial for the producers and promoters of food souvenirs in terms of local tourism development.

## 2. Literature Review

### 2.1. Travel, Shopping Motivation, and Shopper Typology

Shopping is a common leisure activity sought by tourists [24–26]. Shoppers may search for unique commodities and souvenirs and consider brands and logos, products and packaging, prices, product quality, and shop locations [22]. Given that tourism consumption involves experiences from new shopping establishments, tourist pleasure and satisfaction derived from travel are strengthened [17]. Different levels of functional and pleasurable benefits are expected while shopping [27]. Tourists' purchases of souvenirs often constitute a major part of shopping expenditure [20,21]. Souvenirs help

people recall special moments or events during their travel [28]. Moreover, possessing souvenirs is one way to capture the unique qualities of the destination and transport these qualities home as reminders of what made the place special [29].

The phenomenon of shopping while traveling has caught researchers' attention because of the popularity of that activity among tourists. The adventurous nature of travel makes shopping entertaining and enjoyable [30,31]. Kinley et al. [19] explored the push factors of tourist shopping motivations and differentiated between three types of tourists, namely, shopping, experiential, and passive tourists. Among the three types, shopping tourists have the strongest intention of seeking unique local stores, buying special souvenirs, shopping in various stores, or hunting for bargains. Conversely, experiential tourists place emphasis on entertaining experiences, such as relishing a vacation, treating oneself, shopping in various stores, and enjoying interactions with family and friends. Lastly, passive tourists underscore seeking unique local stores, buying special souvenirs, hunting for bargains, and shopping in various stores. Note that passive tourists have the lowest responding scores for most push factors compared with the other tourist types.

Regarding souvenir shopping, Littrell et al. [20] revealed that tourists' souvenir shopping motivation is subject to the attributes of souvenirs, such as the uniqueness of the souvenir itself and whether or not the souvenir can only be purchased locally and represent the locality. Wilkins [23] categorized tourists' souvenir shopping motivations into three types: (1) as presents (buying souvenirs for friends and relatives during holidays or birthdays), (2) as memories (collecting souvenirs for future reminiscing), and (3) as evidence (proof to show that one has been to the destination). The above findings reveal that tourists' souvenir shopping motivations emphasize local uniqueness and affiliation, social enforcement, and travel recollection.

## 2.2. Food and Tourism Development

While traveling, tourists encounter the natural, historical, or sociocultural aspects of a place. The combination of the physical, cultural, and natural environment gives each place its unique appeal or "touristic terroir" and shapes the character of the regional experience [32]. Food is considered not only a basic need, but also a major attraction in some destinations [33]. A study of a rural area confirmed that visitors' interest increases when the locale also offers food and farming events, and when the products offered are consistent with the landscape [34]. Local food has the potential to enhance visitors' experience by connecting them to the culture and heritage of the locality [33]. In addition, most of the pleasure satisfaction of tourists involves dining out and sampling different foods [35]. Traditional foods are essential components in distinguishing the culture of a society and constitute an important medium for cultural expression [36]. Tasting or learning to cook traditional food can also be regarded as an expression of cultural tourism because of tourists' meaningful interaction with host communities [37].

Prior studies have shown that foods provided to vacationing tourists have profound impacts on the economy, culture, and environmental sustainability of destinations; moreover, locally produced products benefit both residents and tourists [38–42]. Tourists' consumption of traditional foods and demands for local foods are associated with their desire for authenticity. The stress on authentic and local produce involves enhancing and creating the iconography of the place [33]. The role of local food as a marker of local identities has been recognized by scholars as placed cultural artifacts, which are often used as a powerful signature of place identity [43]. Therefore, traditional and ethnic foods symbolize culture and identity and have functions of preservation and confirmation [44]. Well-known cuisines have a symbolic status [45]. Tasting food is not only a way to communicate local values, but also serves as a medium to connect visitors to a local distinctive way of life [8]. Thus, foods are full of social, cultural, and symbolic meanings [39] and have a concept of geography and history [37].

For many tourists, the food they enjoy at a destination not only satisfies their appetite, but also becomes intertwined with the identity of a specific destination [46]. Food brands can also potentially develop a dynamic construction of cultural authenticity [47] and are likely to become effective tools

for destination marketing. Local food products have been recognized as a type of souvenir [28,38]. In rural areas, food production and tourism are regarded as possible significant sources of economic development. Using local food as an attraction to tourists increases the circulation of tourist expenditure through the local economy and is beneficial for sustaining local tourism development [7].

*2.3. Motivations for Purchasing Food Souvenirs and Behavioral Intentions*

As food can represent local identities, many tourism destinations are dedicated to producing a local specialty food that caters to tourists' need for souvenirs. The popularity of locally sourced food souvenirs among tourists can impact tourist satisfaction and also benefit the local economy, society, and environment [8,9]. Successful food souvenirs include three essential dimensions: sensory elements, utility aspects, and symbolic meaning [16]. Sensory elements consist of gustatory, visual, and other sensory attributes. Utility aspects cover convenience, health, and natural character. Symbolic meaning consists of authenticity, tradition, and the indigenous aspect. Altintzoglou et al. [48] confirm that taste, quality, authenticity, and local origin are the main factors affecting tourists in choosing and purchasing food as souvenirs.

Previous studies regarding tourists' motivations for souvenir shopping have gained attention among tourism scholars and have generated fruitful results. However, limited research focuses on tourists' motivations for purchasing food souvenirs and their subsequent impact on tourist behavioral intentions. The seminal work of Lin [49] reveals that tourists' motivations for purchasing food souvenirs fall under three groupings: as a gift, as memory reservation, and as travel evidence. The findings of Suhartanto, Dean, Sosianika, and Suhaeni [50] suggest that along with behavioral intentions, uniqueness, authenticity, and taste/value are important determinants of satisfaction for food souvenirs. Furthermore, satisfaction with food souvenirs is a critical driver of tourist satisfaction with visiting the destination.

On the basis of the literature review, the following hypotheses are proposed.

**Hypothesis 1 (H1).** *Online shoppers have different motivational dimensions regarding purchasing food souvenirs.*

**Hypothesis 2 (H2).** *Online food souvenir shoppers with various motivations form different shopper groups.*

**Hypothesis 3 (H3).** *Shopper groups differ in their intentions to search for the information about where food souvenirs originate.*

**Hypothesis 4 (H4).** *Shopper groups differ in their intentions to travel to places where food souvenirs originate.*

**Hypothesis 5 (H5).** *Shopper groups differ in their intentions to recommend places where food souvenirs originate.*

## 3. Research Method

*3.1. Participants*

Qualified participants were aged 18 and above and had purchased food souvenirs online before the time of the survey. The qualifications were stated in the research message. Convenience sampling was used, and participants were self-selected. A total of 410 qualified Internet users participated in the survey, and 402 valid questionnaires were collected.

*3.2. Data Collection*

Results from a survey in Taiwan [51] indicate that Line and Facebook were the two most commonly used social media platforms for shopping. Thus, an online survey was conducted through Facebook and Line for data collection. A research message was sent to these social media users through corresponding social media platforms. Users took the survey voluntarily. Google forms were utilized for questionnaire design and data collection.

### 3.3. Questionnaire

The questionnaire consists of three sections. The first section involves the characteristics of participants, including their gender, age, education level, profession, marital status, average monthly income, average frequency of online purchase of food souvenirs, and average money spent on shopping per month. The second section investigates participants' motivations for purchasing food souvenirs online. The motivational items referenced from Rohm and Swaminathan [13] and Wilkins [23] include 24 questions. Five-point Likert scales were employed for the measurements (1 = "extremely disagree", 5 = "extremely agree"). The third section includes questions on respondents' behavioral intentions after purchasing food souvenirs online, such as willingness to search for information about the place where food specialties were produced, visit such a place, and promote those places. Six questions were provided. Five-point Likert scales were used for measurement (1 = "extremely unlikely" to 5 = "extremely likely").

## 4. Results

The results (Table 1) showed that among the 402 participants, female respondents (68.7%) outnumbered their male counterparts (31.3%). Most participants were aged between 18 and 25 (62.9%), followed by those aged between 26 and 35 (16.5%) and 36 and 45 (11.9%). Most respondents were single (79.6%). A great portion of participants had a college diploma (71.1%), followed by those with only a high school diploma (21.6%). The majority of participants were students (56.2%), followed by homemakers (15.2%). The most common monthly average frequency of food souvenir online purchase was once a month or less (90.8%). Most participants spent 500 new Taiwan dollars (NT$) or less (63.9%), followed by 501–1000 NT$ (25.4%), in terms of monthly average expenditure on food souvenirs online.

A reliability test was performed to investigate participants' consistency in rating 24 motivational questions. Cronbach's $\alpha$ was 0.939, thereby indicating very good consistency. To explore the participants' motivational dimensions of online purchasing of food souvenirs, factor analysis was performed by employing a varimax rotation procedure in the principal component analysis. Five factors with an eigenvalue larger than 1 and factor loadings greater than 0.4 were extracted (Table 2).

The factors are labeled as exploring trends, local affiliation, social interaction, frugal sampling, and enforcing relationships. The variances were 17.56%, 16.29%, 11.70%, 10.25%, and 8.11%, respectively, and the total variance explained was 63.88%. The factor of exploring trends showed that participants are concerned with the popularity of food products and please themselves by purchasing food souvenirs. The factor of local affiliation indicates that participants buy food souvenirs that are related to the sociocultural or natural characteristics of a destination. The factor of social interaction suggested that participants purchase food souvenirs for friends or due to friends' recommendations. The factor of frugal sampling revealed that participants are keen to try new flavors and purchase food souvenirs that are bargains. Finally, the factor of enforcing relationships indicates that participants regard food souvenirs as gifts for friends and relatives and use them to strengthen social relationships. The results reveal that the motivations of online food souvenir shoppers consist of four dimensions. Therefore, H1 is supported.

K-means cluster analysis was employed to classify participants' typologies according to the five dimensions of motivation. The results (Table 3) indicated that participants who purchase food souvenirs online can be classified into four segments: pleasure reminiscers (*n* = 189), journey recallers (*n* = 92), economical tasters (*n* = 70), and social practicers (*n* = 51). Pleasure reminiscers mainly purchase trendy products to please themselves or reminisce about the places where local specialty products originated. The percentage of pleasure reminiscers (47.0%) outnumbered the other three. Journey recallers (22.9%) purchase food souvenirs that connect to specific places. By doing so, they recall their travel experiences and local cultures. Economical tasters (17.4%) are interested in new products that offer opportunities for tasting something new. Social practicers (12.7%) are concerned with increasing social contacts and enforcing interpersonal relationships by giving food souvenirs. The results reveal that four distinct groups of online food souvenir shoppers exist according to the motivations for

purchasing food souvenirs. The largest group consists of pleasure reminiscers, which outweigh the other shopper groups. The second largest group is the journey recallers. Although both groups share the common motive of local affiliation, the former is concerned about the motive of exploring trends, and the latter are not. Economical tasters and social practicers do not concern local connections. The results indicate that four different shopper groups exist among the online food souvenir shoppers based on their shopping motivations. Therefore, H2 is supported.

To investigate whether differences exist between the four types of shoppers in terms of their behavioral intentions, one-way analysis of variance (ANOVA) was performed. The results (Table 4) showed that significant differences exist between the groups in relation to the willingness to search for information (F = 23.29, P = 0.00), willingness to travel (F = 22.67, P = 0.00), and willingness to promote via word-of-mouth (F = 28.73, P = 0.00). Scheffé post hoc analysis was also performed to examine differences among the four groups. The results indicated that pleasure reminiscers showed stronger willingness to search for information, travel, and recommend places where food souvenirs originated than journey recallers, economical tasters, and social practicers. The results indicate that shopper groups differ in their intentions to search for the information, to travel to places, and to recommend places where food souvenirs originate. Therefore, H3, H4, and H5 are supported.

**Table 1.** Respondents' profile.

| Variables | Groups | Frequency | % |
|---|---|---|---|
| Gender | Male | 126 | 31.3 |
| | Female | 276 | 68.7 |
| Age | 18–25 | 253 | 62.9 |
| | 26–35 | 66 | 16.5 |
| | 36–45 | 48 | 11.9 |
| | 46–55 | 31 | 7.7 |
| | 56 and above | 4 | 1 |
| Marital status | Single | 320 | 79.6 |
| | Married | 82 | 20.4 |
| Education level | Junior high and below | 1 | 0.2 |
| | High school | 28 | 7.0 |
| | College | 286 | 71.1 |
| | Graduate school | 87 | 21.6 |
| Occupation | Public employees | 32 | 8.0 |
| | Commercial/industrial employees | 34 | 8.5 |
| | Homemakers | 49 | 15.2 |
| | Students | 226 | 56.2 |
| | Others | 61 | 12.2 |
| Frequency of purchasing food souvenirs per month | 1 time or less | 365 | 90.8 |
| | 2–3 times | 35 | 8.7 |
| | 4–5 times | 1 | 0.2 |
| | 6 times or more | 1 | 0.2 |
| Average monthly expenditure on food souvenirs | 500 NT$ or less | 257 | 63.9 |
| | 501–1000 NT$ | 102 | 25.4 |
| | 1001–2000 NT$ | 36 | 9.0 |
| | 2001–3000 NT$ | 6 | 1.5 |
| | 3001 NT$ or more | 1 | 0.2 |

**Table 2.** Participants' motivational dimensions for purchasing food souvenirs online.

| Motivational Items | Exploring Trends | Local Affiliation | Social Interaction | Frugal Sampling | Enforcing Relationships |
|---|---|---|---|---|---|
| 1. Keeping an eye on the latest news on food souvenirs | 0.750 | 0.209 | 0.153 | 0.098 | 0.138 |
| 2. Making myself feel better when I am not in a good mood | 0.725 | 0.287 | 0.108 | −0.014 | −0.093 |
| 3. Rewarding myself when I accomplish something | 0.687 | 0.297 | 0.143 | 0.093 | −0.105 |
| 4. Having something in common to discuss with my friends | 0.659 | 0.262 | 0.342 | 0.056 | 0.055 |
| 5. Trying trendy food souvenirs | 0.656 | 0.189 | 0.165 | 0.192 | 0.265 |
| 6. Looking for new flavors of food souvenirs | 0.617 | 0.225 | 0.218 | 0.310 | 0.162 |
| 7. A new trial to purchase something via the Internet | 0.434 | 0.291 | 0.024 | 0.130 | 0.021 |
| 8. Reminding me of customs and people of a specific place | 0.248 | 0.810 | 0.111 | 0.167 | 0.196 |
| 9. Reminding me of sceneries in a specific place | 0.280 | 0.789 | 0.167 | 0.165 | 0.121 |
| 10. Connecting myself with a specific place | 0.143 | 0.751 | 0.180 | 0.244 | 0.124 |
| 11. A proof that I have been to a specific place | 0.209 | 0.729 | 0.127 | 0.160 | 0.053 |
| 12. A proof of how well I know local products | 0.310 | 0.729 | 0.190 | 0.127 | −0.017 |
| 13. Reminding me of interesting things during travel | 0.333 | 0.689 | 0.105 | 0.094 | 0.209 |
| 14. Finding appropriate presents for friends | 0.066 | 0.113 | 0.794 | 0.221 | 0.197 |
| 15. Purchasing food souvenirs for friends | 0.407 | 0.263 | 0.645 | −0.066 | 0.007 |
| 16. Because of friends' recommendations | 0.249 | 0.117 | 0.579 | 0.254 | 0.243 |
| 17. Purchasing food souvenirs with friends | 0.395 | 0.154 | 0.549 | 0.270 | 0.033 |
| 18. Being able to taste unique local food | −0.012 | 0.310 | 0.164 | 0.805 | 0.129 |
| 19. Finding food souvenirs that are different from those of my own hometown | 0.275 | 0.333 | 0.127 | 0.570 | 0.179 |
| 20. The price is economical | 0.398 | 0.011 | 0.260 | 0.519 | 0.305 |
| 21. Discounts were offered | 0.467 | −0.025 | 0.196 | 0.475 | 0.284 |
| 22. As presents when visiting friends/relatives | −0.003 | 0.195 | 0.200 | 0.093 | 0.833 |
| 23. Strengthening interpersonal relationships | 0.054 | 0.162 | 0.123 | 0.184 | 0.817 |
| 24. Food souvenirs have special meanings to me | 0.415 | 0.087 | 0.306 | 0.261 | 0.468 |
| **% of variance explained** | **17.56** | **16.29** | **11.70** | **10.25** | **8.11** |

**Table 3.** Typology of online food souvenir shoppers.

| Cluster Name | Cluster 1 | Cluster 2 | Cluster 3 | Cluster 4 |
|---|---|---|---|---|
| Motivational Dimension | Economical Tasters ($n = 70$) | Pleasure Reminiscers ($n = 189$) | Journey Recallers ($n = 92$) | Social Practicers ($n = 51$) |
| Exploring trends | −0.21626 | 0.55261 | −0.39965 | −1.03014 |
| Local affiliation | −1.14565 | 0.41211 | 0.22151 | −0.35435 |
| Social interaction | −0.28772 | 0.15728 | −0.59699 | 0.88897 |
| Frugal sampling | 1.13165 | 0.09418 | −0.57586 | −0.86347 |
| Enforcing relationships | 0.14542 | 0.13096 | −0.78842 | 0.73734 |

**Table 4.** Differences in purchasers of food souvenirs on behavioral intentions.

| Behavioral Intention | Segments | Mean | Scheffé Post Hoc |
|---|---|---|---|
| Searching for information | Economical tasters (1) | 3.22 | (2) > (4)/(1)/(3) |
| | Pleasure reminiscers (2) | 3.87 | |
| | Journey recallers (3) | 3.21 | |
| | Social practicers (4) | 3.24 | |
| Travel | Economical tasters (1) | 3.24 | (2) > (3)/(1)/(4) |
| | Pleasure reminiscers (2) | 3.96 | |
| | Journey recallers (3) | 3.28 | |
| | Social practicers (4) | 3.20 | |
| Promotion via word-of-mouth | Economical tasters (1) | 3.34 | (2) > (1)/(4)/(3) |
| | Pleasure reminiscers (2) | 4.03 | |
| | Journey recallers (3) | 3.13 | |
| | Social practicers (4) | 3.29 | |

## 5. Discussions and Implications

The results of this study indicated that participants' motivations for purchasing food souvenirs online include exploring trends, local affiliation, social interaction, frugal sampling, and enforcing relationships. Motives similar to local affiliation have been identified in past studies on tourists' shopping for souvenirs [20,23]. The motive of enforcing relationships, indicated by purchasing souvenirs as presents, has been acknowledged by past findings [23]. The motives indicating social interaction, exploring trends, and frugal sampling have been recognized in prior research on online shoppers [13]. The outcomes reveal that online food souvenir shoppers exhibit some similar motives to souvenir shoppers in brick-and-mortar settings, as well as to online shoppers of ordinary food products. Note that local affiliation is the motive identified for online and store food souvenir shoppers and for purchasing both food and non-edible souvenirs. Such a phenomenon provides strong support for the uniqueness of souvenirs that symbolize local culture and identity. Conversely, social interaction, exploring trends, and frugal sampling are the motives found for online shoppers who purchase food souvenirs or other types of products. These findings reflect the common motivational natures of general online shoppers.

According to these motivations, participants can be categorized into four consumer segments: pleasure reminiscers, journey recallers, economical tasters, and social practicers. The current results confirm past findings that suggest that consumers with hedonic and utilitarian motivations constitute the two distinct categories of online retail shoppers [52]. Pleasure reminiscers and journey recallers focus on the hedonic value of pleasing themselves or recalling past travel experiences. By contrast, economical tasters and social practicers emphasized the utilitarian value of their purchases. The former is concerned with monetary savings and tasting new products, whereas the latter aim to strengthen their personal relationships with others. The characteristics of pleasure reminiscers and social practicers are revealed through the experiential tourists identified by Kinley et al. [19]. Moreover, the characteristics of journey recallers and economical tasters are highly similar to those of shopping tourists.

The findings revealed that the percentage of pleasure reminiscers outweighed those of the other groups. Most participants were motivated to buy food souvenirs to please themselves or keep up with trends. Such a result conforms to those of previous studies, which indicated that consumers consider shopping a pleasurable and entertaining activity [27,52–54]. Furthermore, the findings confirmed that participants recall their travel experiences by purchasing specific food souvenirs. This outcome is in congruence with previous research that showed that buying food souvenirs is not only a proof of travel, but also generates a collectible item that possesses social and cultural connotations [20,23,28,38,55].

The results of this study also showed that pleasure reminiscers have stronger intentions to search for information about where food souvenirs originated, visit the said places, and recommend these places to friends and relatives compared with journey recallers, economical tasters, and social practicers. This insight may be attributed to the pleasure reminiscers' purchase motivations of local affiliation and exploring trends. The two motives were closely linked to leisure and travel experiences, thereby leading to high intentions to search for information, visit, and recommend destinations to others. Most participants notably consider purchasing food souvenirs online an enjoyable activity. They place emphasis on linkages between food souvenirs and places. The findings indicate that entertainment is a hedonic component of online shopping [11,56,57] and provides consumers with a joyful experience and a means to escape from boredom [8]. Souvenirs also help people remember special moments or events that occurred during travel [28]. Food souvenirs may be a "pull factor" for those who purchase food souvenirs to search for information on destinations, intend to travel, and recommend those places.

Past studies on food souvenir shoppers in an online context are scarce. The results of the current study add to the knowledge of the motivational segments of online food souvenir shoppers and the differences of these segments in travel-related intentions. Given the popularity of e-commerce, the empirical findings are important for researchers in the field of tourism marketing. Moreover, some practical suggestions are provided for the producers of food souvenirs and promoters of local tourism. According to the findings, the needs of pleasure reminiscers deserve great attention because this type of shopper comprises the largest group of online food souvenir shoppers. To attract pleasure reminiscers (whose motives of purchasing food souvenirs are mainly for exploring trends and local affiliation), products should reveal local characteristics to draw attention and facilitate a connection. Pictures of local landscapes or unique features can be printed on the packaging of food souvenirs. Products should also emphasize local ingredients and the production method. The shapes of products can mimic iconic local features so as to render them visually impressive and interesting. Moreover, creating new flavors for food souvenirs that keep pace with market trends can make the products appealing and competitive. To appeal to journey recallers, the second largest group of online food souvenir shoppers with motives focusing on local affiliation, products should show strong cultural elements on packaging and emphasize local ingredients and traditional production methods. Printed material depicting local anecdotes or imagery can be included in the packaging to strengthen the local image and even make the products collectible items. Suggestions are also made for the other two groups of online food souvenir shoppers, which comprise approximately 30% of the respondents. To attract economical tasters, who are frugal shoppers and interested in tasting new products, unique flavors of food souvenirs can be produced to induce their desire to purchase, and discounts can be offered on special occasions. Conversely, to draw social practicers, who place importance on social interactions and use food souvenirs to enforce their relationships, the packaging design should facilitate a social function. The packaging should be visually attractive. In addition, the packaging should provide handles for easy transport, and small cards can be attached to the packaging for writing blessings.

## 6. Conclusions

This study has certain limitations. The first limitation is the high percentage of young participants. This situation arose because of the tendency of online surveys to have young participants. In Taiwan, Generation Z (12–23 year olds) has the highest rate of Internet usage at 99.8% [58]. However, Generation Z's main purposes for using the Internet include instant messaging, searching for news and information, accessing movies and music, and community forums. The frequency of their online shopping and money spent for online shopping are lower than those of the rest of the age groups [57]. This phenomenon is noteworthy. Young consumers are the most frequent Internet users, and yet they are the least sufficient monetarily. Thus, marketing strategies should focus more on providing eye-catching imagery of food souvenirs to quickly attract customers' attention while they search on the Internet. At the time they are financially available, they are likely to purchase food souvenirs online. Furthermore, the study participants are limited to those who purchased food souvenirs online. Some of the respondents may

have been to places where food souvenirs originate, whereas others have not. The participants' past travel experiences are unknown, and their travel-related intentions may be affected by those events. Finally, future studies can explore the effects of participants' past travel experiences on purchasing food souvenirs online and their travel-related intentions. Finally, the sociocultural and economic functions of food souvenirs in affecting the sustainability of local tourism can be investigated.

**Author Contributions:** Conceptualization, S.-C.L.H. and C.-Y.W.; methodology, S.-C.L.H. and Y.-R.Y.; formal analysis, S.-C.L.H. and Y.-R.Y.; investigation, S.-C.L.H. and Y.-R.Y.; resources, S.-C.L.H. and C.-Y.W.; data curation, S.-C.L.H.; writing—original draft preparation, S.-C.L.H. and C.-Y.W.; writing—review and editing, S.-C.L.H. and C.-Y.W.; project administration, S.-C.L.H. All authors have read and agreed to the published version of the manuscript.

**Funding:** This research received no external funding.

**Conflicts of Interest:** The authors declare no conflict of interest.

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
