# Peer review of "Motivational Typology of Online Food Souvenir Shoppers and Their Travel-Related Intentions"

_sustainability, doi:10.3390/su12187624_

Round 1

Reviewer 1 Report

Please briefly address previous Comments and mark the text through tack Changes to see the modification.    Comments and Suggestions for Authors   introduction   the authors fail to explain the background how tourist considers souvenir as commercial goods or a local product. In addition, introduction lacks clarification what authors refer to the commercial or local product? What is the difference between them?

Afterwards, authors state the purposes of study which are not consistent with the problem statement. Since the first purpose is to investigate the consumers’ motivations to purchase online souvenirs and their segmentation based upon their level of motivations. However, the introduction does not present any information regarding the segmentation of the customers. The second purpose is to examine the differences between segmented consumers; however, the second problem is just inquiring the individuals’ interest of tourism instead of exploring the differences between consumers.        

Research Method

The authors failed to add the appropriate reasoning about the selection of respondents and the research process. It is better to explain how the internet users who purchased food souvenirs online were selected for this study. Describe the process of mail-survey for data collection and processing rather to add the name of Facebook and line only. Questionnaire is missing

Results

On line 156, please add appropriate reason why the authors select loadings greater than 0.4 for factor analysis. The author must discuss first the significance of K-means cluster analysis applied in this study. And how it can better classify the participants. How the participants who purchase food souvenirs online are classified into four segments? Motivation segments as demonstrated in study purpose and appropriate discussion related to four segments is missing in the study.

 Discussions and Suggestions

Authors must address the theoretical contribution of the study in more detail. As most of the respondents are students and their frequency of purchase food souvenirs is 1 or less. How did you justify that the results of this large sample is appropriate for the practical implications. Please add conclusion section separately.

Author Response

The authors thank the reviewer for his/her valuable comments on the manuscript. For specific responses to the comments, please see the attached file.

Reviewer 2 Report

Results and discussions are too general.

Author Response

(The authors gave the same response as above.)

Reviewer 3 Report

The manuscript "Motivational Typology of Online Food Souvenir Shoppers and Their Travel-related Intentions" gives interesting insights into a narrow topic. According to the study's findings “there are four types of souvenir online buyers; pleasure reminiscers (47.0%), journey recallers (22.9%), economical tasters (17.4%), and social practitioners (12.7%). Pleasure reminiscers have stronger intentions on searching information of, traveling to, and recommending the places where food souvenirs originate compared with the other three types of consumers”. Unfortunately, considering the narrowness of the topic and the methodological limitations, I feel that the paper is not suitable for publication.

Considering the venue of publication, I think it is necessary to highlight the contribution of this study to the literature on sustainability. This is required mainly in the introduction and in the future research sections.

The paper does not demonstrate an adequate understanding of the relevant literature in the field and cites a limited range of literature sources.

Please add a section focusing on “motivations for purchasing food souvenirs”. Your study is missing literature and contemporary studies from the field of “motivations for purchasing food souvenirs”. While you briefly referring to “motivations for purchasing food souvenirs” literature in the introduction and the literature review sections your study is lacking a concrete approach to it.

There are several significant papers that have ignored. I am listing some citations from various databases for the author(s)’ consideration. Some of these are too recent to have been included in the writing of the paper. I am not suggesting that these are necessary for revising the paper.  I simply encourage the author(s) to examine them in case they will be useful.

Pabian, A., Pabian, A. and Brzeziński, A., 2020. Young People Collecting Natural Souvenirs: A Perspective of Sustainability and Marketing. Sustainability, 12(2), p.514.

Ansari, F., Jeong, Y., Putri, I.A. and Kim, S.I., 2019. Sociopsychological aspects of butterfly souvenir purchasing behavior at Bantimurung Bulusaraung National Park in Indonesia. Sustainability, 11(6), p.1789.

Katsikari, C., Hatzithomas, L., Fotiadis, T. and Folinas, D., 2020. Push and Pull Travel Motivation: Segmentation of the Greek Market for Social Media Marketing in Tourism. Sustainability, 12(11), p.4770.

Pham, Q.T., Tran, X.P., Misra, S., Maskeliūnas, R. and Damaševičius, R., 2018. Relationship between convenience, perceived value, and repurchase intention in online shopping in Vietnam. Sustainability, 10(1), p.156.

Ijaz, M.F. and Rhee, J., 2018. Constituents and consequences of Online-shopping in Sustainable E-Business: An experimental study of Online-Shopping Malls. Sustainability, 10(10), p.3756.

Swanson, K.K. and Timothy, D.J., 2012. Souvenirs: Icons of meaning, commercialization and commoditization. Tourism Management, 33(3), pp.489-499.

Anastasiadou, C. and Vettese, S., 2019. “From souvenirs to 3D printed souvenirs”. Exploring the capabilities of additive manufacturing technologies in (re)-framing tourist souvenirs. Tourism Management, 71, pp.428-442.

Paraskevaidis, P. and Andriotis, K., 2015. Values of souvenirs as commodities. Tourism Management, 48, pp.1-10.

Collins-Kreiner, N. and Zins, Y., 2011. Tourists and souvenirs: changes through time, space and meaning. Journal of Heritage Tourism, 6(1), pp.17-27.

There is not a separate section for the hypotheses development. In my opinion, the part of the hypotheses development is the most important section of the theoretical background section. The author(s) should focus on this part and develop it in-depth, Furthermore, the theoretical rationale for each hypothesis needs to be strong and grounded in the literature.

Please add more details on the methodology section.  Describe the sampling process and the sample more detailed. Please meet points like: How were the participants approached? From what population were you sampling? Etc.). Please further elaborate on the use of snowball or convenience sampling methods. What were the concrete questions asked (measures) (not only for motives)? What was the setting of the study?  I think the author(s) also need to justify why the research used Taiwanese participants (especially considering it was an online experiment) and how they are universally representative.

The discussion section mainly restates the findings rather than giving a thoughtful discussion of the study’s implications for theorists and practitioners. What does this study add to previous knowledge?

It would be nice if the author(s) could discuss more limitations and future research propositions of this work.

Moreover, it seems that the paper should be checked and corrected by a professional editor in order for grammatical and syntactic mistakes to be avoided.

While I was interested in reading this research, I think the author(s) have a lot of work ahead of them before this research is publishable.

Author Response

(The authors gave the same response as above.)

Round 2

Reviewer 2 Report

The author propose five hypothesis, but after that there is no reference to them.The explanations of the results are ambiguous, sometimes contradictory (for example line 247-248) and far too general.

Author Response

The author propose five hypothesis, but after that there is no reference to them.The explanations of the results are ambiguous, sometimes contradictory (for example line 247-248) and far too general.

Response: The sentence is revised. Please see Lines 246-248. The references to the five hypotheses are provided. Please see Lines 230-231, Lines 248-250, & Lines 261-263.

Reviewer 3 Report

The revision now effectively addresses all of my concerns. The authors have done a nice job of responding to my earlier comments/suggestions. In my view, the paper has noticeably improved and now presents a tighter story. 

Author Response

The revision now effectively addresses all of my concerns. The authors have done a nice job of responding to my earlier comments/suggestions. In my view, the paper has noticeably improved and now presents a tighter story.

Response: The authors want to thank the reviewer’s valuable comments on the manuscript.

Round 3

Reviewer 2 Report

The authors improved the content of the scientific paper for a better presentation of the research.

This manuscript is a resubmission of an earlier submission. The following is a list of the peer review reports and author responses from that submission.

Round 1

Reviewer 1 Report

1. This study primarily focuses on tourists’ shopping behaviours and motivations. I wonder how this article may fit with the scope of the “Sustainability” journal. I would suggest the authors focus more on “sustainable development.”

2. To frame this study into a sustainable context, I would recommend the authors to read the following literature.

Sims, R. (2009). Food, place, and authenticity: local food and the sustainable tourism experience. Journal of Sustainable Tourism, 17(3): 321-336. Everett, S. and Aitchison, C. (2008). The role of food tourism in sustaining regional identity: A case study of Cornwall, South West England. Journal of Sustainable Tourism, 16(2): 150-167. Hjalager, A.-M., and Johansen, P.H. (2013). Food tourism in protected areas – sustainability for producers, the environment and tourism? Journal of Sustainable Tourism, 21(3): 417-433. Hall, M., Sharples, L., Mitchell, R., Macionis, N., and Cambourne, B. (2004). Food Tourism Around the World, (Eds.), Routledge, New York: NY.

3. There are some unique characteristics of food souvenirs. The authors should talk about the differences between food souvenirs and other gift souvenirs (e.g. craft). In many cases, food is considered as fast-consuming products, particularly snacks and beverages. It may not be perceived or treated as memorabilia – “collecting souvenirs for future reminiscing” (p. 2, line 87-88) is questionable. In addition, some people are “food neophobia.” They may not even enjoy nor appreciate the local food products. For example, some people do not like durian or its by-products (e.g., durian candies, cake) due to its unique or strong taste/smell. It’s depending on what type of food.

4. The motivations could be very different if the travelers purchase food souvenirs for themselves versus purchase for others. However, the authors did not explain the differences between these two types of situations and buying motives.

5. Page 3, line 100-101: “Local food has the potential to enhance visitors’ experience by connecting them to the culture and heritage of the locality.” I wonder if the local food and local culture/heritage are always positively correlated. Would it be possible that some travelers like the culture and history of a country but dislike the local food/cuisine? If they are not always positively correlated, it would be worthwhile to ask the question “why” to unearth the underlying reasons and differences.

6. Page 3, line 111-113: “Prior studies have shown that foods provided to vacationing tourists have profound impacts on the economy, culture, and environmental sustainability of destinations, and locally produced products benefit both residents and tourists.” The authors fail to explain and elaborate on this point – e.g. How does it create profound impacts on environmental sustainability? I would suggest that the authors should dedicate at least one section to talk about the relationship between food souvenirs and environmental sustainability. By doing this, it will make this study fits well with the scope of the “Sustainability” journal.

7. The authors should develop some hypotheses for empirical testing.

8. Buying food online is not the same experience as buying food during a trip or vacation. In a similar vein, the motives for buying souvenirs from aboard or domestic may not be the same. It would be helpful if the authors can provide more information about the differences.

9. The authors should further explain the relationships between 4 clusters and 5 motivational dimensions.

10. What measuring instruments were used to measure “willingness to search for information,” “willingness to travel” and “willingness to promote via word-of-mouth”? It would be helpful if the authors can clarify and explain.

11. Page 8, line 216-218: “The finding is in congruence with previous research that showed that buying food souvenirs is not only a proof of travel but also a collectible item that possesses social and cultural connotations [15,18,23,30,45].” Many people collect craft souvenirs but not food. I wonder why food souvenirs are considered as collectible items. Do travelers collect food souvenirs without consuming them? Or do they collect food packages (e.g. coca-cola, bottle) or photographs?

12. I’m afraid to say this study does not provide enough new insights nor theoretical and practical implications. It seems to me that this paper is more suitable for tourism and hospitality journals than sustainable and ecological journals.

Reviewer 2 Report

Introduction

The authors have started with the economic benefits of tourism industry then present the comparison of Taiwanese and Japanese food souvenirs, and move on to discuss the convenience and monetary advantages of online souvenirs purchasing, which are completely irrelevant from original problem of this study.  Moreover, the authors fail to explain the background how tourist considers souvenir as commercial goods or a local product. In addition, introduction lacks clarification what authors refer to the commercial or local product? What is the difference between them?

There is a lack of literature that leads to the proper problem formulation. Another concern arises, when authors investigate the second issue regarding the individuals’ interest to visit the places where souvenirs originate. It creates confusion since former and latter are two distinct problems. Former relates to the purchase of souvenirs online and latter tries to explore the individuals’ interest of tourism. How come authors combine these two issues together? Afterwards, authors state the purposes of study which are not consistent with the problem statement. Since the first purpose is to investigate the consumers’ motivations to purchase online souvenirs and their segmentation based upon their level of motivations. However, the introduction does not present any information regarding the segmentation of the customers. The second purpose is to examine the differences between segmented consumers; however, the second problem is just inquiring the individuals’ interest of tourism instead of exploring the differences between consumers.        

Literature Review

2.1. Discusses a lot about shopping souvenirs but referring to the purpose of this study, examining the motivations of tourists to buy food souvenirs online. I was unable to find any literature explaining the online purchase of food souvenirs and behavioral intention.

2.2. Explaining how local food connects tourists with family, strangers and friends. I don’t know why authors are trying to describe this correlation. If it is not related with the problem statement and purpose of this study. In summary, 2.1 and 2.2 are irrelevant in the context of present study, which cannot disintegrates the literature with the problem and purpose of this study. 

Research Method

The authors failed to add the appropriate reasoning about the selection of respondents and the research process. It is better to explain how the internet users who purchased food souvenirs online were selected for this study. Describe the process of mail-survey for data collection and processing rather to add the name of Facebook and line only. Questionnaire is missing

Results

On line 156, please add appropriate reason why the authors select loadings greater than 0.4 for factor analysis. The author must discuss first the significance of K-means cluster analysis applied in this study. And how it can better classify the participants. How the participants who purchase food souvenirs online are classified into four segments? Motivation segments as demonstrated in study purpose and appropriate discussion related to four segments is missing in the study.

 Discussions and Suggestions

Authors must address the theoretical contribution of the study in more detail. As most of the respondents are students and their frequency of purchase food souvenirs is 1 or less. How did you justify that the results of this large sample is appropriate for the practical implications. Please add conclusion section separately.